# Anthracene-Induced Alterations in Liverwort Architecture In Vitro: Potential for Bioindication of Environmental Pollution

**DOI:** 10.3390/plants13152060

**Published:** 2024-07-26

**Authors:** Maya Svriz, Cristian D. Torres, Lucas Mongiat, Elisabet Aranda, Nahuel Spinedi, Sebastian Fracchia, José Martín Scervino

**Affiliations:** 1Institute of Research in Biodiversity and Environment (INIBIOMA), CONICET-UNCo, San Carlos de Bariloche 8400, Argentina; mayasvriz@gmail.com (M.S.); torresc@comahue-conicet.gob.ar (C.D.T.); naspinedi@comahue-conicet.gob.ar (N.S.); 2Departamento de Física Médica, Centro Atómico Bariloche, Comisión Nacional de Energía Atómica, San Carlos de Bariloche 8400, Argentina; lucas.mongiat@cab.cnea.gov.ar; 3Consejo Nacional de Investigaciones Científicas y Técnicas, Ciudad Autónoma de Buenos Aires 1425, Argentina; 4Department of Microbiology, Farmacy Faculty and Institute of Water Research, University of Granada, Ramón y Cajal, Bldg. Fray Luis 4, 18071 Granada, Spain; earanda@ugr.es; 5Instituto de Micología y Botánica (INMIBO), Facultad de Ciencias Exactas y Naturales, Universidad de Buenos Aires, Ciudad Autónoma de Buenos Aires 1428, Argentina; sebafrac@yahoo.com.ar

**Keywords:** bryophytes, liverwort, pollution, anthracene, bioindicator, morpho-architectural traits, conservation

## Abstract

Polycyclic aromatic hydrocarbons (PAHs) are widespread globally, primarily due to long-term anthropogenic pollution sources. Since PAHs tend to accumulate in soil sediments, liverwort plants, such as *Lunularia cruciata*, are susceptible to their adverse effects, making them good models for bioindicators. The aim of this study was to probe the impact of anthracene, a three-ring linear PAH, on the growth parameters of *L. cruciata* and the relationship established with the internalization of the pollutant throughout the phenology of the plant. Intrinsic plant responses, isolated from external factors, were assessed in vitro. *L. cruciata* absorbed anthracene from the culture medium, and its bioaccumulation was monitored throughout the entire process, from the gemma germination stage to the development of the adult plant, over a total period of 60 days. Consequently, plants exposed to concentrations higher than 50 μM anthracene, decreased the growth area of the thallus, the biomass and number of tips. Moreover, anthracene also impinged on plant symmetry. This concentration represented the maximum limit of bioaccumulation in the tissues. This study provides the first evidence that architectural variables in liverwort plants are suitable parameters for their use as bioindicators of PAHs.

## 1. Introduction

Polycyclic aromatic hydrocarbons (PAHs) are organic compounds and chemical pollutants included in the priority list (EPA, Environmental Protection Agency, US) that arise from human activities such as the combustion of crude oil, fossil fuels, and the burning of industrial and household waste [1]. The release of chemical pollutants dates back to the Industrial Revolution, but it has significantly accelerated over the past fifty years in terms of both the rate of release and how far they spread in the environment [2]. Roughly 90% of PAHs present in the environment are bound to soil particles, with the greatest concentrations found in industrial areas. In contrast, lower concentrations of PAHs are found in urban and rural soils [3]. The PAHs can be accumulated and persist for a long time, leading to changes in the soil’s microbial activity and nutrient cycling [4]. Its negative effects can also result in decreased soil fertility and increased soil erosion [5,6]. Furthermore, in soil with clay content, the biodegradation of PAHs is limited [7]. Anthracene, the most basic form of PAHs in terms of structure, can be found in coal tar and it serves as a starting material for the manufacture of synthetic fibers, insecticides, several dyes such as alizarin or coating materials, among others [8,9]. Is also likely to be found with other PAHs in asphalts and creosote [10]. Particularly, anthracene is a PAH with the chemical formula C_14_H_10_. It is a solid, colorless compound that is soluble in organic solvents such as benzene and chloroform, but insoluble in water. It is commonly used as a starting material in the production of dyes and inks due to its intense blue fluorescence [11,12]. However, anthracene is a toxic contaminant that can negatively affect plant growth and soil quality [13,14].

Exposure to high levels of anthracene can lead to reduced plant growth, decreased seed germination, and altered root morphology [15,16]. It can also interfere with plant photosynthesis and nutrient uptake, which can further affect the overall plant health [17,18,19]. Anthracene has been shown to have negative effects on the growth and development of liverwort plants such as *Marchantia polymorpha* and *L. cruciata* [20,21,22]. In studies by Spinedi, Rojas et al. [20] and Spinedi, Storb et al. [21] exposure to anthracene resulted in reduced growth of *M. polymorpha* gametophytes and altered chlorophyll content. Additionally, anthracene exposure resulted in increased lipid peroxidation and oxidative stress in *M. polymorpha*, indicating cellular damage. In another study by Storb et al. [22], anthracene exposure caused a significant decrease in *L. cruciata* growth and altered the plant’s physiological and biochemical processes. Specifically, anthracene exposure led to a decrease in photosynthetic pigments, altering the plant growth. These findings suggest that this contaminant has the potential to cause significant harm to liverwort plants. However, the spectrum and mechanisms of action in other related liverworts, essential components of many ecosystems, remain unexplored.

*Lunularia cruciata*, like other bryophytes, has a haploid gametophytic thallus, with radial growth as a product of dichotomous branching, which exhibits a unique pattern of repeated forking that creates a Y-shaped structure [23]. In this pattern, the axis of the liverwort plant repeatedly forks into two equal branches [24]. By growing in this manner, the branching pattern produced from a meristem cell located at the apex of the thallus allows liverworts to efficiently capture light and maximize their surface area, which enables them to effectively capture nutrients from the surrounding environment. However, Streubel et al. [24] demonstrated that *M. polymorpha*, in response to shade evasion, can inhibit the activity of its meristems through gene regulation. This inhibition results in an asymmetrical branching, affecting the whole architecture of the thallus. Additionally, liverworts have the ability to regenerate from each branch point, allowing them to continue growing and reproducing even after being damaged. One possible hypothesis that we put forward in this work is that exposure of *L. cruciata* to anthracene will disrupt the functioning of the meristem cells, leading to alterations in the plant’s dichotomous growth pattern. Specifically, anthracene will interfere with cell division and differentiation, resulting in stunted growth, abnormal branching, and a reduced photosynthetic surface. These changes will ultimately have negative consequences for the health and growth of *L. cruciata* and may affect the overall structure and function of the ecosystem in which it resides.

It is known that bryophytes are frequently employed as model plants in research due to their capacity to react to environmental pollution [25]. Due to its morphology, their thallus can absorb all nutrients and contaminant [26]. For this reason, they are excellent bioindicators for monitoring contaminants [27,28]. Bioindicators, while less precise than physicochemical techniques, offer advantages such as capturing the cumulative effects of pollutants and habitat changes over time, reflecting indirect biological impacts, and providing effective ecosystem-level monitoring—all at a lower cost and with less time investment [29]. Furthermore, bioindicators possess several advantages: they can encapsulate the combined impacts of chemical pollutants and habitat alterations over the lifespan or residency of an organism, reflecting the indirect biological repercussions of pollutants. Additionally, biotic indices serve as a potent tool for ecosystem monitoring, offering insights into changes at the ecosystem level [29]. In this sense, there are multiple bioindicators for environmental monitoring; however, it is important to establish new, quick, and simple methods for the identification of contaminants such as PAHs. Bryophytes, as one of the most popular bioindicator taxa, hold particular importance due to their rapid growth, easy detection, and economic feasibility for study as a bioindicator of these characteristics [30]. Despite these advantages, there remains a gap in identifying a bioindicator specifically for hydrocarbon pollution. In this context, it is interesting to identify the way in which the responses of plants exposed to contaminants are reflected in morpho-architectural traits, which could improve the use of certain species as bioindicators. It is crucial to monitor and control the levels of certain contaminants such as anthracene in the environment to prevent its adverse effects on plant growth and ecosystems. We propose that studying of morpho-architectural traits in liverwort plants present in contaminated environments will help identify the evident impacts of anthracene on plant growth and architecture and reinforce the potential use of plant morphological traits as bioindicators for environmental monitoring. In order to provide tools in this regard, in this study we assess morphological indicators for the exposure of *L. cruciata* plants to anthracene.

## 2. Results

### 2.1. Gemma Germination

The frequency curves of germinated gemmae were similar and showed overlapping between treatments (Figure 1).

### 2.2. Plant Growth and Morpho-Architectural Traits

From day 50 onwards, a reduction in the area of the thallus exposed to 100 and 250 μM anthracene was evident (*p <* 0.001); no significant differences were observed in this regard between the control plant and those of the 50 μM treatment (*p* = 0.664; Figure 2A). Similarly, plants from the 100 and 250 μM treatments presented significantly lower biomass (79–90%) than the controls (*p <* 0.001). On average, the biomass of plants in the 50 μM treatment was almost half of that of the control plants, although this difference was not significant (Figure 2B).

The thallus diameter differed between the *L. cruciata* plants that grew in the different treatments; the lowest values corresponded to the plants from treatments 100 μM and 250 μM (Table 1). The number of nodes reached did not differ significantly among treatments. In response to the treatment with the lowest concentration of anthracene (50 μM), the plants showed a significantly lower number of tips compared to the control plants (Table 1). The reduction in the number of tips was even greater for 100 and 250 μM treatments, which had half the tips of the control plants (Table 1). The plants corresponding to the control and 50 µM treatments presented the lowest symmetry indices and were similar to each other, while for treatments 100 µM and 250 µM, the values were higher. The ratio between the thallus diameter and the number of tips observed displayed the same trend as the symmetry index (Table 1). The plants with a symmetrical branching pattern were those of the control and 50 μM treatments (Figure 3A,C), while the asymmetric plants were those of the 100 μM and 250 μM treatments (Figure 3B,C).

### 2.3. Evaluation of Anthracene over Time

Regardless of the exposure time, a significant incorporation of anthracene was observed in the plant tissues in respect to the control treatment (Figure 4). In all treatments, as the days progressed, a noticeable reduction in the concentration of anthracene in the culture medium was observed. Conversely, the plants exhibited an increase in the concentration of anthracene in their tissues over the same period (Figure 5). The uptake of anthracene by the plants occurred from day 10 of exposure in all treatments and the maximum concentration of anthracene incorporated into the thallus was close to 50 µM (Figure 5). Plants exposed to 50 µM of anthracene removed all the anthracene from the medium after 50 days of exposure (Figure 5A). Plants exposed to 100 µM bioaccumulated 50% of the anthracene from the medium after 60 days of exposure (Figure 5B), while this capacity was reduced by 75% after exposure to 250 µM anthracene (Figure 5C).

## 3. Discussion

In this study, we evaluate the changes induced by anthracene in the morpho-architecture of liverworts in vitro, thereby assessing their viability as a tool for monitoring environmental pollution. The variation in the growth and morpho-architectural traits of *L. cruciata* plants grown under different concentrations of the pollutant anthracene provided information on their tolerance levels and bioaccumulation capacity in environments with different degrees of contamination. In our results, the gemma germination was similar in all the treatments applied, indicating that anthracene does not delay or affect gemmae germination. These results are consistent with findings for *M. polymorpha*, where anthracene concentrations did not modify the final germination of gemmae in relation to the control treatment [20]. Additionally, the experiments agree with those of Carginale et al. [31], who observed that the germination of propagules of *M. polymorpha* has resistance to contaminants. This suggests that relying on gemmae germination as a bioindicator of toxicity may be inadequate due to the inherent resistance of this species, which suggests low resolution of this parameter as a general bioindicator of toxicity.

The presence of anthracene in high concentrations had a negative effect on the growth of the *L. cruciata* plants from day 50 onwards. This was reflected in a decrease in both the thallus area and its biomass. This coincides with the internalization of the compound in tissues plants. These results clearly show that the presence of anthracene at concentrations higher than 100 μM causes significant reduction in the growth of *L. cruciata*, which could be evident from 50 days after exposure to this compound. Our results align with the evidence presented by other authors, where it is apparent that one of the most frequent responses of plants to environmental stressors involves alterations in growth and reduction in biomass, particularly in the presence of contaminants such as heavy metals [32], pesticides [33], or PAHs [34,35,36].

Anthracene affected not only growth but also architecture, most notably with exposures to 100 and 250 μM. The decrease in the number of tips on the thallus in the highest concentration treatments reflects negative effects on the meristematic activity responsible for branching. However, anthracene exposure did not affect the maximum branching order reached. The data suggests that anthracene exposure did not reduce the number of dichotomies but may have affected the growth of branches after meristem division. This is supported by the increase in the symmetry index for 100 and 250 μM treatments, indicating a possible imbalance between growth and branching. Control plants had a symmetry index close to 1, which demonstrates that, in the absence of anthracene, branching occurs mainly through dichotomies. Plants that grew with anthracene concentrations greater than 100 μM presented symmetry indices close to 2.3, showing a significant reduction in the number of apices in relation to the number of nodes. Furthermore, the ratio between the diameter of the thallus and the number of tips increases with anthracene concentration, suggesting imbalances between growth and branching. A recent study in *M. polymorpha* demonstrated that the development of bifurcations after meristem duplication is regulated by light, specifically through the activity of clade III SPL genes [24]. The activity of this gene determines that under shade conditions, just one of the daughter meristems produced by dichotomy has the capacity to outgrow while the other one remains dormant. Considering that the plants were under the same light conditions during our experiment, it is possible that anthracene impacts a specific point in the pathway regulating genes responsible for the dormancy or activity of meristems after duplication. To test this hypothesis, future research should examine the expression levels of these genes under varying anthracene concentrations. Taking all the results together, the observed changes in plant architecture provide valuable insights into the effects of anthracene exposure. Our findings suggest that certain concentrations of anthracene lead to smaller plants, reduced bifurcation capacity after meristem duplication, and a lower number of branches relative to their total diameter. Future studies should focus on identifying the molecular pathways involved and how anthracene specifically influences these developmental processes. Understanding these mechanisms will help clarify the broader impacts of anthracene on plant growth and architecture.

In the present work we observed that the reduction of anthracene content in the culture medium was concomitant with the bioaccumulation in plants. The bioaccumulation of anthracene occurred from day 10 after the essay began. Several studies have observed that the increase in concentration of PAHs in the tissues of various plant species is closely linked to the levels of exposure to the contaminant [37,38,39]. Only the plants exposed to the lowest concentration of anthracene were able to completely phytoextract the contaminant from the culture medium. This capacity progressively decreased as the anthracene concentration increased. The reduction in phytoextraction from the medium with increasing anthracene concentration may be attributed to the stress induced in the plant by high concentrations of contaminants. In the presence of low concentrations of contaminants, [40] observed that a species of green alga, *Ulva lactuca*, was able to extract 98% of the anthracene from the culture medium (5 µM) through the activation of antioxidant enzymes. Furthermore, these antioxidant mechanisms were also identified in the liverwort *M. polymorpha* [21]. Thus, it is conceivable that plants exposed to 50 µM of anthracene could employ similar stress mitigation strategies as observed in those studies. The percentages of bioaccumulation for *L. cruciata* in treatments 100 µM and 250 µM were 50 and 25% respectively, similar to the bioaccumulation pattern in vascular plants, where at high concentrations, the percentages of anthracene accumulation fluctuate between 20% and 50% [41,42].

## 4. Conclusions

The results obtained indicate that anthracene significantly affects the morpho-architectural traits of *L. cruciata*, encompassing parameters such as area, biomass, and thallus diameter, as well as the number of apices, symmetry index, and diameter/tips observed ratio, all of which reflect evident symptoms of the presence of this contaminant. The variability observed in these traits, both at low and high concentrations of the contaminant, underscores the potential use of morpho-architectural traits as biological indicators of soil pollution. Additionally, we highlight its ability to incorporate anthracene in its tissues and reduce its concentration in the environment, making it an efficient bioaccumulator and positioning it as a potential tool in phytoremediation strategies for environmental pollution. This is the first study to date that assesses and proposes morpho-architectural traits in liverwort plants as bioindicators of contaminated soils, offering a novel and simple method for detecting this type of pollution.

## 5. Materials and Methods

### 5.1. Biological Material

Gemmae or propagules of *L. cruciate* plants were extracted from gemma cups, to obtain samples from independent individuals that remained homogeneous throughout the experiment. Gemma cups came from mother plants of *L. cruciata* in axenic culture conditions. These plants were growth in Petri dishes of 90 mm of diameter containing minimum medium [43] in a growth chamber with a 16/8 h (light/dark) photoperiod, at a temperature of 24 °C, and at a constant light intensity of 60 μM photons m^−2^ s^−1^ in controlled conditions.

### 5.2. Chemicals and Reagents

Anthracene (97% HPLC grade) was purchased from Sigma-Aldrich, St. Louis, MO, USA. All the solvents employed were of high-performance liquid chromatography (HPLC) grade. Specifically, acetonitrile and HPLC water were provided by PanReac (AppliChem, Barcelona, Spain), hexane and acetone by WWR Chemicals (International Eurolab, Barcelona, Spain), and phosphoric acid by Fisher Chemical (Madrid, Spain). Additionally, all other chemicals and reagents utilized were of analytical grade or higher purity.

### 5.3. Gemma Germination

Samples of 25 gemmae from gemma cups of mother plants of *L. cruciata* were placed into plastic Petri dishes containing minimum medium [43] with four different anthracene concentrations. Anthracene was added from a stock solution diluted in acetone (5 mM). Culture medium was prepared adding the stock solution immediately after autoclaving, to reach the final concentrations of 0, 50, 100, and 250 μM of anthracene [44]. To avoid potential solvent toxicity, the medium was stirred under sterile conditions for 15 min to facilitate the evaporation of acetone. The experimental methodology employed has been extensively validated [42,44], although crystal formation may be observed [20]. Control treatments included a minimum medium lacking anthracene (0 µM) and a medium containing only acetone (anthracene-free). Twelve replicates were used for each treatment. The growing conditions for the gemmae were maintained identically to those of the mother plants. Gemmae germination was recorded every 8 h until the last gemma germinated. The gemmae that presented rhizoids were considered germinated.

### 5.4. Plant Growth and Morpho-Architectural Traits

To test plant growth, 16 germinated gemmae from each treatment (64 in total) in the previous experiment were transferred into new Petri dishes with the same treatments, leaving one gemma per plate. The culture conditions were consistent with those explained above. In each plant photographs were taken every 10 days to estimate the area of the thallus with ImageJ 1.54g software (https://imagej.nih.gov/ij/) accessed on 2 November 2023. After 60 days, the samples were dried in an oven at 70 °C until constant weight to evaluate the biomass of each individual. From the images (64 plants pictures) obtained on day 60, in addition to the measurement of the thallus area, the following traits were measured: diameter of the thallus (with ImageJ Software), maximum sequence of nodes and total number of tips. “Node” refers to the point that gives rise to a dichotomy with either one or two branches as a product (Figure 3A,B). The maximum sequence of nodes (which is equal to the maximum branching order achieved) was considered as the largest sequence of consecutive nodes observed in each thallus (Figure 3A,B). Therefore, to obtain an approximation of the thallus symmetry, we calculated a hypothetical tips number expected if branching were completely symmetric. In a completely symmetric thallus, the sequence of nodes would be the same in every possible sense, or in other words, all tips within the thallus would represent the same branching order (Figure 3A). Therefore, when considering the maximum sequence of nodes of each thallus, the expected number of tips after complete symmetric branching was calculated using the following equation:$$expected\;n{}^{\circ}\;tips=2*2^{\max n{}^{\circ}\;nodes}$$

The symmetry index was then calculated as the following ratio: n° expected tips/n° observed tips. This index would take the value of 1 in completely symmetric thallus, while higher values imply greater asymmetry in the branching pattern. On the other hand, we calculated the ratio between the diameter of the thallus and the number of tips observed to evaluate the effect of the treatments on the relation between growth and branching.

### 5.5. Evaluation of Anthracene over Time

The timing and amount of anthracene internalized by *L. cruciata* were determined in an assay over time. Gemmae (obtained from the gemma cups of mother plants) were transferred to Petri dishes and subjected to the same treatments and conditions as in the previous tests. Enough gemmae were planted to enable the collection of individuals every 10 days over a 60-day period, aiming to achieve the required dry mass (30 mg per sample across three replicates) for contaminant quantification. We utilized a minimum of 100 gemmae per replicate to facilitate harvesting within 10 days, reducing to 50 gemmae within 20 days, and 10 gemmae within 30 to 60 days per treatment. In order to determine the dissipation of the contaminant in the culture media, a portion of the culture medium was taken for analysis at each harvest time. The concentration of anthracene in the thallus dry mass and culture medium (free-tally control and anthracene treatments) was analyzed using the methodology of [44]. Anthracene was extracted with n-hexane/acetone (2:1 *v*/*v*) through three sonication cycles (15 min at 60 °C), evaporated with a N_2_ flow, and resuspended in acetonitrile. Samples were measured using an Agilent 1200 HPLC (Agilent, Technologies, Palo Alto, CA, USA) equipped with a DAD detector and using a column C18 Synergy Fusion (80 Å; 4 μM, 4.6 × 150 mm; Phenomenex^®^, Madrid, Spain). Compounds were separated using water + 0.1% phosphoric acid (A) and acetonitrile (B) as eluent buffers in isocratic mode and a flow rate of 1 mL/min. A calibration curve with the pure standards was used for the quantification and the results were expressed as μM. In addition, the presence of anthracene in 60-day old plants was observed via fluorescence microscopy (Olympus BX51, Tokyo, Japan) as described previously [20,32]. Wavelengths were chosen based on the excitation and emission spectra of anthracene 370–470 nm [45].

### 5.6. Data Analysis

Repeated measures analyses (ANOVAR) were used to evaluate the thallus area, between treatments throughout the experiment. At the end of the experiment, biomass and final diameter of the thallus were compared between treatments by means of linear models. When necessary, non-homogeneous variance models were considered, using the VarIdent function [46]. Architectural traits were compared between the thallus exposed to each treatment and the control. The symmetry index and the diameter/n° tips ratio was analyzed using linear models, while for the count variables (n° maximum sequence of nodes and n° apices), models where applied that assume Poisson error distribution with function logarithmic. All pairwise multiple comparisons were performed using Tukey tests in the package.

## Figures and Tables

**Figure 1 plants-13-02060-f001:**
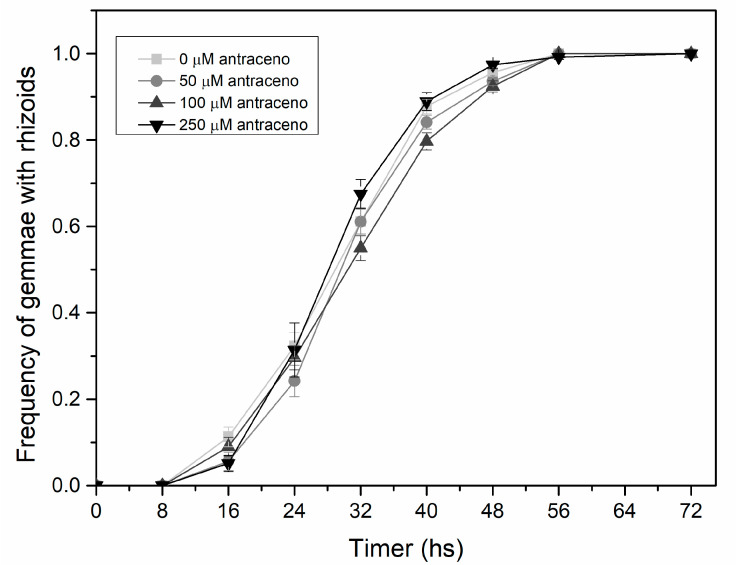
Frequency of *L. cruciata* gemmae with rhizoids every 8 h at different levels of anthracene concentrations, 0, 50, 100, and 250 µΜ.

**Figure 2 plants-13-02060-f002:**
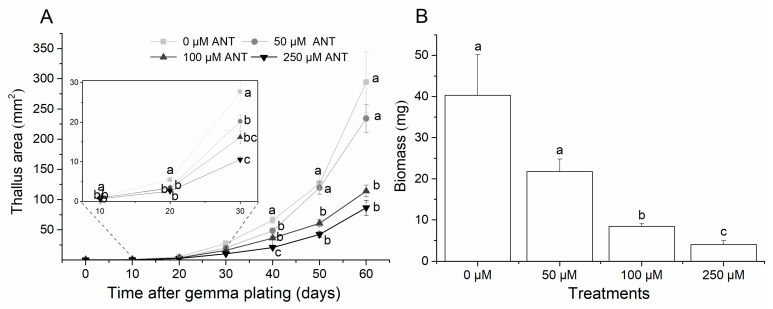
(**A**) Mean values (±SE) of the growth area of *L. cruciata* measured every 10 days for treatments 0, 50, 100, and 250 µΜ. The subframe represents the thallus area from 0 to 30 days. (**B**) Mean values (±SE) of accumulated biomass of *L. cruciata* measured on day 70 for treatments 0, 50, 100, and 250 µΜ. Statistical differences between treatments are indicated with letters on top of the bars (Tukey paired-contrasts test, *p <* 0.05).

**Figure 3 plants-13-02060-f003:**
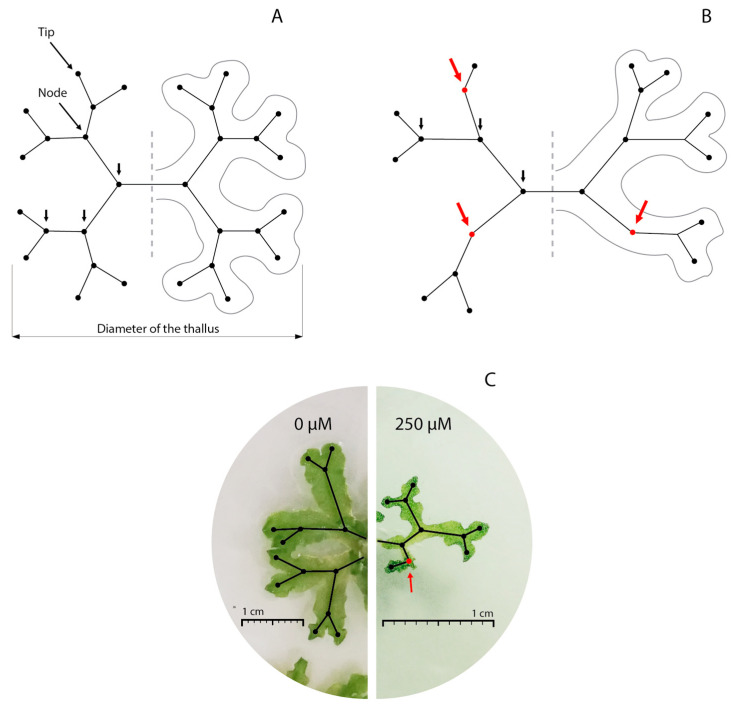
*Lunularia cruciata* thallus branching patterns. Black lines indicate branches, black dots represent nodes and tips (meristems). (**A**) Hypothetical representation of a completely symmetric thallus. (**B**) Asymmetric thallus branching in presence of contaminant. Short black arrows taken together indicate the maximum sequence of nodes. Red arrows and red dots indicate the absence of dichotomy (simple branching). (**C**) Growth and branching of thallus in 0 µM and 250 µM treatment.

**Figure 4 plants-13-02060-f004:**
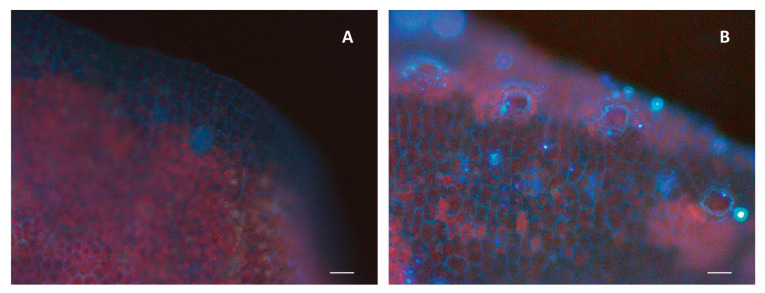
Fluorescence micrograph of a control plant (**A**) and a plant exposed to 250 µM of anthracene (**B**). Scale bars = 10 µm. Differences in the blue light intensity indicate the presence of anthracene.

**Figure 5 plants-13-02060-f005:**
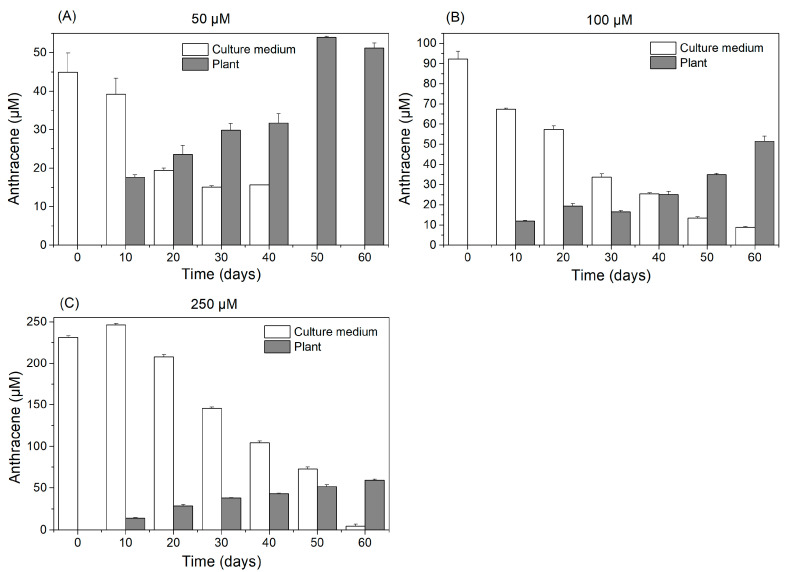
Mean values (±SE) of the anthracene concentration (μM) present at 60 days in the culture medium and in *L. cruciata* plants grown in the different anthracene treatments: (**A**) 50 μM, (**B**) 100 μM, and (**C**) 250 μM.

**Table 1 plants-13-02060-t001:** Architectural variables measured in *L. cruciata* plants for the 0, 50, 100, and 250 μM anthracene treatments. Different lowercase letters indicate significant differences between treatments.

Variables	Treatments				*p*-Value
	0 µM	50 µM	100 µM	250 µM	
Diameter (mm)	33.15 ± 2.61 a	27.85 ± 1.84 ab	21.56 ± 1.43 bc	20.53 ± 1.35 c	<0.001
Number reached nodes	5.22 ± 0.14	4.87 ± 0.12	4.37 ± 0.26	4.2 ± 0.13	0.734
Number of tips observed	29.33 ± 2.10 a	21.50 ± 1.05 b	14.37 ± 0.88 c	14.60 ± 1.09 c	<0.001
Symmetry index	1.27 ± 0.18 a	1.51 ± 0.08 a	2.29 ± 0.17 b	2.30 ± 0.17 b	<0.001
Diameter/tips observed	1.13 ± 0.06 a	1.32 ± 0.12 ab	1.51 ± 0.07 b	1.43 ± 0.07 b	0.027

## Data Availability

Data are contained within the article.

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
