# Peer review of "Anthracene-Induced Alterations in Liverwort Architecture In Vitro: Potential for Bioindication of Environmental Pollution"

_plants, 2024, doi:10.3390/plants13152060_

Round 1

Reviewer 1 Report

Comments and Suggestions for Authors

This is an interesting research work and written well. Therefore I would like to recommand to accept without further revision.

Author Response

Response to reviewer 1_

We want to express our deepest gratitude for the time and effort you have dedicated to reviewing our manuscript.

Best regards

Dr. Scervino J. Martín

Reviewer 2 Report

Comments and Suggestions for Authors

The article deals with a specific small issue, in particular, the impact anthracene-induced alterations in Lunularia cruciata. The authors correctly and in detail describe the work performed. The results obtained are rather interesting and according to the authors, theoretically Lunularia cruciata can be used as a bioindicator.

Some small comments are in the att. text. One general comment concerns the citation of sources. In the text, when mentioning a link with the preposition “by”..... it is still necessary to put the last name of at least one of authors, and then give the number of the link in square brackets. E.g. Line 60-61 instead “In studies by  [20,21] exposure to anthracene....” should be changed for “in studies by Spinedi, Rojas  et al. and Spinedi, Storb et al.  [20,21]

I don't understand what the authors mean by the diameter of the thallus? In taxonomy of liverwotrs such feature as diameter of the thallus is nevere used. 

Author Response

Response to reviewer 2_

We want to express our sincere gratitude for the time and effort dedicated to reviewing our manuscript. In reference to your comment, we agree that thallus diameter is not a taxonomic characteristic to use. In our study, we decided to use "thallus diameter" as a variable that can be affected by contamination issues. This is related to the type of growth exhibited by the plant, which in the case of L. cruciata is radial.

Below we detail point by point the changes made in the text. We have corrected the reference according to your recommendation in the text:

We have corrected the references of lines 60-61 according to your comments. We changed “In studies by  [20,21] exposure to anthracene.... for “in studies by Spinedi, Rojas  et al. and Spinedi, Storb et al.  [20,21]

References were also corrected in the Lines 64, 78 and 173. We add “Storb et al., [22], Streubel et al., [24] and carginale et al., [31] respectively.

We have changed stem for thallus (Line 227)

We hope that these revisions improve the clarity of our manuscript

Best regards

Dr. Scervino J. Martín

Reviewer 3 Report

Comments and Suggestions for Authors

This paper reports the results of experiments on the effects of  anthracene on the liverwort Lunularia . As might be expected anthracene has profound effects which are clearly documented . However, I have major issues with this paper : in my view the results are badly overinterpreted and the text is far too long. The paper would be much clearer  if the results and discussion were separated and a lot of the speculation removed. As it stands it is difficult to see clearly the most significant findings in the work and separating these from the speculatory comments.

Comments on the Quality of English Language

Generally satisfactory

Author Response

Response to reviewer 3_

Thank you for your feedback on our manuscript. We appreciate your insights and have carefully considered your suggestions for improvement.

In response to your comments, we have made significant revisions to the manuscript to address the issues raised. Specifically, we have rewritten the section of results and discussion to enhance clarity and removed speculative comments to ensure our focus remains solely on the observed findings. While we appreciate the suggestion to separate the discussion and results sections, it is worth noting that the decision to keep them integrated was made based on the feedback from the other two reviewers and the editor, who found this structure to be appropriate for the manuscript.

Below we give you more details of the rewritten paragraphs:

  1. Introduction

Lines 100-102 and 111-114

  1. Results and discussion

2.1. Gemma germination

We have rewritten the paragraph according to your comments Lines 124-127

2.2. Plant growth and morpho-architectural traits

We have rewritten the paragraph according to your comments Lines 158-161;168-170 and 173-183

  1. 3. Evaluation of anthracene over time

We have rewritten the paragraph according to your comments Lines 199-201; 208-209

We believe these changes have greatly improved the manuscript by making the significant findings more prominent and removing any potential ambiguity introduced by speculative statements.

We hope that these revisions meet your expectations and improve the clarity of our work. Thank you once again for your valuable feedback, which has undoubtedly strengthened our manuscript.

Sincerely

Dr. Scervino J. Martín

Reviewer 4 Report

Comments and Suggestions for Authors

A good ECOTOXICOLOGY study. 

However, your title should be on experimental laboratory controlled condition study. 

You have not proven it to be effective and it's efficacy  in the field based condition. 

So, please revise your title accordingly. I could not think for you. 

As a researcher, always reserve the importance on the field based reseacher to be recommended in the future. 

Hope my comments click your ECOTOXICOLOGY understanding and a room for future study. 

Good luck, Sir. 

Author Response

Comment: A good ECOTOXICOLOGY study.

However, your title should be on experimental laboratory controlled condition study. You have not proven it to be effective and it's efficacy in the field based condition. So, please revise your title accordingly. I could not think for you.  As a researcher, always reserve the importance on the field based reseacher to be recommended in the future.  Hope my comments click your ECOTOXICOLOGY understanding and a room for future study.

Thank you for your valuable feedback on our manuscript.  In response to your comment, we acknowledge the importance of distinguishing fieldwork from laboratory work. Therefore, we have included the term "in vitro" in the title to clearly indicate the laboratory-based nature of our study.

Best regards

Dr. Scervino J. Martin
